# Cholesterol Sulfate: Pathophysiological Implications and Potential Therapeutics

**DOI:** 10.3390/biom15050646

**Published:** 2025-04-30

**Authors:** Xiaoqian Yu, Siman Lei, Ying Shen, Tao Liu, Jun Li, Jia Wang, Zhiguang Su

**Affiliations:** 1Center for High Altitude Medicine and Department of Pain Management, State Key Laboratory of Biotherapy, West China Hospital, Sichuan University, 1 Keyuan 4th Road, Gaopeng Street, Chengdu 610041, China; yxq@wchscu.edu.cn (X.Y.);; 2College of Life Science, South China Agricultural University, Guangzhou 510642, China

**Keywords:** cholesterol, sterol, sulfotransferase, metabolism, signal transduction, keratinocyte, Alzheimer’s disease, cancer, immune response, gut microbiota

## Abstract

Cholesterol sulfate (CS) is a naturally occurring cholesterol derivative that is widely distributed across various tissues and body fluids. In humans, its biosynthesis is primarily mediated by the sulfotransferase (SULT) 2B1b (SULT2B1b). Over the years, CS has been found to play critical roles in various physiological processes, including epidermal cell adhesion, sperm capacitation, platelet adhesion, coagulation, glucolipid metabolism, bone metabolism, gut microbiota metabolism, neurosteroid biosynthesis, T-cell receptor signaling, and immune cell migration. In this review, we first introduce the endogenous regulation of CS biosynthesis and metabolism. We then highlight current advances in the understanding of the physiological roles of CS. Finally, we delve into the implications of CS in various diseases, with a particular focus on its mechanism of action and potential therapeutic applications. A comprehensive understanding of CS’s physiological function, biosynthesis regulation, and role as a disease modifier offers novel insights that could pave the way for innovative therapeutic strategies targeting a wide range of conditions.

## 1. Introduction

Steroids are generally conjugated with other molecules, such as amino acids, proteins, and sugars, to prolong their biological time in the organism and facilitate the transport of steroids to target sites [1]. In contrast to hundreds of synthetic steroid conjugates, only two types of steroid conjugates occur in living organisms, namely, those conjugated either by sulfotransferases (SULTs) to form sulfates or by uridine diphosphate-glycosyltransferases (UGTs) to form glucuronides. The hydrophilicity of sulfates and glucuronides facilitates the excretion of conjugated steroids in urine and bile. In contrast to glucuronidation, which mainly controls the levels and biological activity of unconjugated hormonal steroids, steroid sulfates possess their own biological activities that differ from those of the respective unconjugated steroids. For example, although both steroid sulfates and their unconjugated steroids can bind to neurotransmitter receptors, their binding sites are different, and their effects on ligand-gated ion channels are often opposite (positive vs. negative) [2]. Moreover, the concentration of many sulfated steroids in blood is usually several-fold higher than that of their unconjugated counterparts, such as DHEA and estrone [3].

Cholesterol sulfate is among the most abundant steroid sulfates in circulation, and its concentration in human blood (platelets and red blood cells) ranges from 1.3 to 2.6 μg/mL [4]. In addition to being a predominant circulating sterol sulfate, CS has also emerged as a significant lipid constituent in a variety of biological fluids and tissues, such as seminal plasma, urine, bile, skin, adrenal glands, the uterine endometrium, and intestinal, kidney, and liver tissue [5]. As a hydrophilic excretion form of cholesterol, CS not only modulates cholesterol homeostasis by targeting cholesterol synthesis and esterification pathways but also has various physiologic functions through interactions with extracellular matrix proteins or cellular adhesive receptors. In addition to having the most investigated physiologic roles in epidermal keratinocyte differentiation and biological membrane homeostasis, CS also acts as a key player in glucolipid metabolism, the immune response, gut microbiota metabolism, and neurosteroid synthesis [5]. It is now evident that CS is associated with various diseases, ranging from diabetes and ulcerative colitis to cancer, bone metabolic disorders, skin diseases, and Alzheimer’s disease. In this review, we aim to elucidate this interesting molecule and focus mainly on the biological pathways influencing human health and disease.

## 2. CS Biosynthesis and Metabolism

CS is synthesized in a variety of tissues, such as the epidermis, red blood cells, and platelets. The production rate of CS has not been precisely determined, but it is estimated to be approximately 45 mg/day, comparable to that of dehydroepiandrosterone (DHEA) sulfate in the bloodstream [4]. The biosynthesis of CS involves two main steps: the activation of inorganic sulfate and the sulfonation of cholesterol mediated by sulfotransferases (SULTs) (Figure 1). Dietary-derived sulfate, an essential micronutrient, undergoes intestinal absorption through specific sulfate transporters. Following cellular uptake, sulfate activation initiates through ATP sulfurylase (ATPS) activity within the bifunctional 3′-phosphoadenosine 5′-phosphosulfate (PAPS) synthase complex [6]. This enzymatic reaction transfers the adenosine monophosphate (AMP) moiety from ATP to sulfate, generating adenosine-5′-phosphosulfate (APS), characterized by a high-energy phosphoric–sulfuric acid anhydride bond crucial for sulfate activation [7]. Subsequently, the APS kinase (APSK) domain of PAPS synthase phosphorylates APS at the 3′-position to yield the universal sulfate donor PAPS. The final sulfonation step is mediated by sulfotransferases (SULTs) that transfer the activated sulfate group from PAPS to the 3β-hydroxyl group of cholesterol, ultimately producing CS [8].

SULTs are widely distributed in human tissues, such as the skin, liver, lungs, brain, prostate, placenta, mammary glands, platelets, and kidneys. SULTs are located mainly in the cytoplasm and the Golgi membrane [9]. Cytoplasmic SULTs are involved in the metabolism of endogenous substances (such as hormones and neurotransmitters) and exogenous substances (such as drugs), whereas membrane-bound SULTs mainly participate in the sulfation of tyrosine residues [10]. The human *SULT* gene family mainly includes *SULT1*, *SULT2*, *SULT4*, and *SULT6*, among which *SULT2* is involved primarily in the sulfation of hydroxysteroids. The human SULT2 family comprises two subfamilies, SULT2A1 and SULT2B1. SULT2A1 preferentially uses DHEA as a substrate and is, therefore, also referred to as DHEA sulfotransferase [11]. The SULT2B1 subfamily consists of *SULT2B1a* and *SULT2B1b*, which are produced from the same gene, *SULT2B1*, on chromosome 19q13.3 through alternative splicing. Typically, *SULT2B1b* is expressed at higher levels than *SULT2B1a*. Although both SULT2B1a and SULT2B1b can sulfate DHEA, SULT2B1b preferentially catalyzes the sulfation of cholesterol to form CS, whereas SULT2B1a tends to sulfate pregnenolone [12,13].

CS can be excreted through pathways such as skin shedding, urine, and feces. Both rodents and humans exhibit high levels of CS in urine, intestinal tissues, and feces, with CS in feces constituting 2~20% of total cholesterol [4]. Intracellular CS can be hydrolyzed by the steroid sulfatase gene (*STS*)-encoded steroid sulfatases (SSases), which are located on the endoplasmic reticulum or Golgi membranes, into free cholesterol. The liberated cholesterol can then bind with steroidogenic acute regulatory protein (StAR), increasing the half-life of the StAR protein. This process promotes cholesterol transport into mitochondria and enhances the synthesis of steroids and high-density lipoprotein (HDL) [14]. The human *STS* gene is located on Xp22.3-Xpter and is expressed in various tissues, including the placenta, mammary glands, skin, lungs, ovaries, adrenal glands, and brain. *STS* is associated with high levels of estrogen and androgen within tumors, thereby contributing to the growth of steroid hormone-dependent cancers, such as breast cancer, ovarian cancer, prostate cancer, and endometrial cancer [15].

## 3. Physiological Functions

CS has diverse physiological functions in the human body. It participates in epidermal differentiation and barrier function formation, modulates T-cell receptor signaling, regulates glucose and lipid metabolism, exerts neuroprotective effects, maintains cell membrane stability, and is involved in platelet adhesion and coagulation processes.

### 3.1. CS Regulates Keratinocyte Differentiation and Skin Development

The epidermis is composed of the innermost basal layer, spinous layer, granular layer, and outermost stratum corneum, each of which is characterized by a particular type of keratinocyte [16]. After initiating differentiation in the basal layer, keratinocytes progressively move upward toward the surface of the epidermis during the differentiation process, forming a characteristic hierarchical structure. The protective function of the skin is primarily maintained by the stratum corneum, which is composed of multiple layers of lipid matrix interspersed with keratinocytes. The lipids in the stratum corneum mainly include equal amounts of ceramides, free fatty acids, and cholesterol. This balanced lipid composition is crucial for the barrier function of the stratum corneum. CS is distributed in a gradient manner across different epidermal layers, accounting for 1%, 5%, and 1% of the total lipids in the basal layer, granular layer, and stratum corneum, respectively [17]. Compared with the nonpolar and hydrophobic lipid matrix, the acidic sulfate groups of CS impart strong polarity and hydrogen bonding capacity, thereby affecting the fundamental structure of the lipid matrix. Computer modeling of the 3D structure of the lipid layers in the stratum corneum revealed that these layers are very rigid, with each lipid layer containing only one or two water molecules [18]. The incorporation of the polar molecule CS into the lipid layers can loosen the lipid packing and increase lipid hydration, significantly increasing the permeability of the stratum corneum [19].

In addition to serving as a component in the lipid matrix in the stratum corneum, CS also regulates keratinocyte differentiation through multiple mechanisms, maintaining the homeostasis of the epidermal barrier. (1) CS regulates the transcription of genes encoding cornified envelope proteins, such as transglutaminase 1 (TGM1) and involucrin, by coordinating with the AP-1 transcription factor to increase gene expression [20]. (2) CS is more effective than phosphatidylserine and phorbol esters in activating the η isoform of protein kinase C (PKCη), which catalyzes the phosphorylation of epidermal structural proteins and increases the formation of the cornified envelope. PKCη can also phosphorylate and activate AP-1, thereby increasing the transcription of differentiation-associated proteins, such as TG-1 and involucrin [21,22]. (3) During keratinocyte differentiation, the production of keratins stops after the cells move from the basal layer to the granular layer and instead begin to produce various barrier-forming proteins, such as filaggrin and loricrin. CS can induce filaggrin expression either by inducing the expression of Retinoic acid-related orphan receptor alpha (*RORα*) or serving as a ligand for *RORα* [23]. Additionally, CS is closely related to normal skin desquamation [17]. An increase in CS enhances cell membrane stability and intercellular cohesion in the stratum corneum, leading to disruption in the desquamation process and skin lesions characterized by dryness and scaling [24,25].

### 3.2. CS, as a Ligand, Plays Fundamental Roles in Signaling Pathways

RORα is a widely distributed transcription factor that regulates gene expression and is involved in various physiological functions, such as carbohydrate and lipid metabolism, cell cycle regulation, development, and tumorigenesis [26]. Although both CS and cholesterol can serve as endogenous ligands for RORα and participate in RORα-associated molecular signaling pathways, the sulfate group in the CS molecule forms more hydrogen bonds with the ligand-binding domain of RORα than the 3-hydroxyl group in the cholesterol molecule does, thereby enhancing molecular interactions [27]. Consequently, CS has a much greater affinity for RORα than does cholesterol.

CS plays a significant role in the substrate specificity of phosphoinositide 3-kinase (PI3K) [28]. PI3K can be activated by insulin, the platelet-derived growth factor (PDGF), and other factors. Activated PI3K phosphorylates phosphatidylinositol 4,5-bisphosphate (PI-4,5-P2) to generate the second messenger phosphatidylinositol 3,4,5-trisphosphate (PI-3,4,5-P3), which recruits signaling proteins from the cytoplasm to the cell membrane, thereby activating a series of signaling pathways that regulate physiological functions, such as carbohydrate and lipid metabolism, protein synthesis, and cell proliferation. In addition to PI-4,5-P2, PI3K can also phosphorylate phosphatidylinositol (PI) and phosphatidylinositol monophosphate (PI-4-P) [29]. CS enhances the substrate selectivity of PI3K for PI-4,5-P2, thereby maintaining the normal signaling pathway of the second messenger PI-3,4,5-P3 [28].

### 3.3. CS Maintains Biological Membrane Homeostasis

CS is an important component in various types of biological membranes, such as those in red blood cells, platelets, and sperm, where it acts as a stabilizer (Figure 2). The hydrophilicity of the sulfate group and hydrophobic side chain of cholesterol enable CS to have the ability to stabilize biological membranes and inhibit membrane fusion. CS interacts with phospholipids in a manner similar to that of cholesterol, but the intermembrane exchange rate of CS is significantly faster than that of cholesterol [30], which allows for the sorting of hydrocarbons in the lipid bilayer and stabilizes the membrane structure.

In the bloodstream, physiological concentrations of CS can increase the stability of the red blood cell membrane, protecting the cells from osmotic hemolysis [4]. The exposure of red blood cells to high temperatures (51 °C) induces membrane rupture and hemolysis, whereas preincubation of the cells with CS can partially protect them against rupture [31]. Enveloped animal viruses, such as Sendai virus, require fusion between the viral envelope and the cytoplasmic membrane to enter cells [31,32]. CS, rather than cholesterol, can inhibit fusion between the virus and red blood cells, thereby suppressing Sendai virus-induced hemolysis [31]. CS is also a major component in platelets, and the concentration of CS in normal rat or human platelets ranges from 164 to 512 pmol/mL, accounting for approximately 1% of the total cell content [33]. The sulfate groups and steroid ring structure of CS increase its affinity for platelets. CS can be directly integrated into the platelet membrane without altering the platelet cholesterol composition to increase thrombin-induced production of thromboxane B2, which is an end product of arachidonic acid metabolism, leading to an increase in arachidonic acid-induced platelet aggregation and serotonin secretion [34].

CS exists at relatively high concentrations in the male reproductive tract, where it is absorbed by the sperm and is primarily localized in the acrosomal membrane [35]. CS hydrolysis in the sperm membrane may lead to membrane destabilization and trigger sperm motility, which are essential for sperm capacitation. During sperm maturation in the epididymis, CS acts as a membrane stabilizer and an effective inhibitor of the acrosomal protease. CS has a significant effect on sperm quality, with higher levels of CS in the semen of oligospermic or infertile patients, making it a critical predictor of semen quality [36]. In the female reproductive tract, CS is hydrolyzed by steroid sulfatase, which can reverse the inhibitory effects of CS on sperm function, leading to the acrosome reaction and a series of events that culminate in sperm capacitation and fertilization [37,38,39].

Entamoeba histolytica is a kind of amoeboid protozoan parasite, and its inhabitation in the human gastrointestinal tract causes amoebiasis, a global public health issue. Entamoeba infection and transmission occur mainly through the oral ingestion of its cysts [40]. CS is found in amoebic cysts, where it decreases membrane permeability and maintains the spherical shape of the membrane, attenuating the phagocytic activity of amoebic cysts and partially disrupting the transmission of Entamoeba histolytica [41]. Therefore, CS may offer a new strategy for the prevention and treatment of parasitic diseases, but the specific molecular mechanisms underlying this effect require further investigation.

### 3.4. CS Modulates Inflammatory and Immune Responses

Inflammation is a critical part of an immune system’s response to harmful stimuli, involving various inflammatory mediators and cell types [42]. 5-lipoxygenase (5-LO) is the key enzyme responsible for generating leukotrienes and other soluble inflammatory mediators [43]. Upon Ca^2+^ influx into the cell, 5-LO binds to the cell membrane and converts arachidonic acid into biologically active leukotrienes, which induce severe inflammation in asthma and bronchitis [44]. CS, as a component of the cell membrane, can directly interact with 5-LO, weakening its interactions with the membrane and thereby reducing leukotriene synthesis and release, ultimately inhibiting the inflammatory response [45]. CS has been shown to suppress IL-17 production in T cells, directly affecting osteoclastogenesis and inhibiting the inflammatory response in rheumatoid arthritis [46]. In addition, CS can inhibit proinflammatory macrophage polarization by increasing nicotinamide adenine dinucleotide phosphate (NADPH) levels and activating the AMP-activated protein kinase (AMPK)-cAMP response element-binding protein (CREB) signaling pathway, alleviating ischemic stroke in a mouse model [47].

The T-cell surface antigen receptor (TCR) is a transmembrane complex composed of antigen-binding subunits (TCRαβ) and CD3 signaling subunits (CD3ζζ, CD3δε, and CD3γε). The TCR is essential for T-cell activation, regulation, and function [48]. Upon binding to peptide antigens presented by major histocompatibility complex (MHC) molecules, the TCR is activated via tyrosine phosphorylation of the CD3 subunits, initiating a cascade of molecular events, including adapter protein phosphorylation, signaling complex formation, intracellular calcium flux, immunological synapse formation, cytokine secretion, and cell proliferation. Cholesterol can directly bind to the TCRβ transmembrane (TCRβ-TM) region to enhance TCR affinity for peptide–MHC ligands [49]. However, CS binds to the TCRβ-TM region with an affinity 300-fold greater than that of cholesterol [50], competitively disrupting TCR–ligand complex formation and inhibiting TCR signaling. CS can also suppress T-cell activation by inhibiting the tyrosine phosphorylation of the CD3 subunit. Furthermore, CS can induce dysfunctional T-cell microvilli, which are crucial for TCR signal transduction and T-cell function [51,52]. Additionally, CS acts as an intrinsic mediator in T-cell signaling, regulating thymocyte development. CS-deficient mice exhibit increased sensitivity to self-antigens, whereas CS injection into the thymus can inhibit thymic selection [53].

Dedicator of cytokinesis protein 2 (DOCK2) can activate the small G protein Rac by mediating the guanine nucleotide exchange, thereby stimulating lymphocyte activation, differentiation, trafficking, and homing [54]. CS, rather than cholesterol or other cholesterol derivatives, directly binds to the catalytic DHR2 domain of DOCK2, inhibiting its guanine-exchange activity and greatly suppressing DOCK2-mediated Rac activation and lymphocyte migration [55]. The ability of CS to suppress immune cell migration is exemplified in the tear film, where CS mediates immune evasion in the eyes [55]. Mice with a specific knockout of the *Sult2b1* gene develop severe UV- or antigen-induced ocular surface inflammation, which can be reversed by the administration of CS-containing drops [55]. Moreover, the macrophage-inducible C-type lectin receptor (Mincle) is an innate immune receptor involved in allergic skin inflammation [56]. CS serves as an endogenous ligand for Mincle to modulate allergic skin inflammation [56], although the underlying molecular mechanisms remain unclear.

### 3.5. CS Regulates Glucose and Lipid Metabolism

CS orchestrates systemic glucose homeostasis through dual regulatory mechanisms. On one hand, CS inhibits hepatic gluconeogenesis by modulating the activity of hepatocyte nuclear factor 4α (HNF4α), a liver-specific transcription factor that governs the expression of key gluconeogenic genes, including *PEPCK* (Phosphoenolpyruvate carboxykinase) and *G6Pase* (glucose-6-phosphatase G-6-pase). The activity of HNF4α is positively correlated with its acetylation level. CS can inhibit HNF4α acetylation and interfere with its nuclear transport, thereby suppressing hepatic gluconeogenesis [57,58]. Genetic evidence from *Sult2b1* knockout models confirms this regulatory axis. *Sult2b1* knockdown decreases hepatic CS synthesis and subsequently inhibits the expression of the deacetylase sirtuin 1. This leads to an increase in HNF4α acetylation and a subsequent enhancement of HNF4α’s gluconeogenic activity [58]. On the other hand, CS enhances glucose-stimulated insulin secretion in pancreatic β-cells by optimizing mitochondrial bioenergetics. Insulin secretion relies on mitochondrial function and ATP production within β-cells. CS maintains the integrity of the mitochondrial membrane structure and enhances the efficiency of oxidative phosphorylation in β-cells, thereby increasing ATP production and promoting insulin secretion [6].

CS is a key regulator of lipid metabolism, exerting its effects through multiple mechanisms. CS acts as an endogenous ligand for RORα, which in turn activates AMPK. This activation leads to the suppression of hepatic liver X receptor α (LXRα), thereby inhibiting the expression of lipogenic genes. This cascade of events significantly reduces hepatic lipid accumulation [59]. Furthermore, CS attenuates cholesterol biosynthesis by inhibiting 3-hydroxy-3-methylglutaryl coenzyme A (HMG-CoA) reductase, which is the rate-limiting enzyme in cholesterol synthesis. It also blocks cholesterol esterification by inhibiting lecithin/cholesterol acyltransferase (LCAT), further reducing cholesterol levels [60]. In addition, CS serves as a major substrate for the synthesis of pregnenolone sulfate in adrenal glands or gonads, where the highly active mitochondrial cytochrome P450 cholesterol side chain cleavage enzyme (P450 SCC) converts CS to pregnenolone sulfate [61], which is a crucial step in steroid hormone production.

### 3.6. CS Alters Gut Microbiota-Derived Metabolites

CS is closely related to the gut microbiota and host intestinal metabolism. Compared with native starch (NS), dietary resistant starch (RS) is less digestible by intestinal enzymes but can be efficiently utilized by the gut microbiota [62]. Moreover, RS can significantly alter the gut microbiota composition, increasing the abundance of Bacteroides, Akkermansia, and Bifidobacteria, while decreasing the abundance of Firmicutes. These changes in microbial composition lead to alterations in host metabolic components, such as CS levels. For example, alterations in the gut microbiota in obese individuals have been associated with significantly elevated CS levels in both plasma and feces [63]. Recent studies have shown that the implantation of Bacteroides thetaiotaomicron into the cecum of germ-free mice increases the host’s CS levels [64,65]. Mechanistically, the genome of Bacteroides thetaiotaomicron encodes steroid *SULT*, APS kinase, and AS. Once sulfate is transported into bacterial cells, AS catalyzes the conversion of ATP and sulfate to APS, which is then phosphorylated by APS kinase to produce PAPS. Finally, *SULTs* catalyze the transfer of the sulfate group from PAPS to cholesterol to form CS [65]. Furthermore, CS affects host cell responses, reducing leukocyte migration in a dose-dependent manner, potentially exerting a negative effect on immune cell migration [65,66]. However, the mechanisms by which CS influences the composition and gene expression of the gut microbiota are not fully understood.

### 3.7. CS Regulates Neurosteroid Synthesis and Affects Nervous System Function

CS is also found in brain tissue, with the cerebellum having the highest CS content [67]. The brain contains many steroid hormones and sulfotransferase enzymes, such as P450SCC, P450-17 hydroxylase, P450 aromatase, 3β-hydroxysteroid dehydrogenase, sulfotransferases, sulfatases, and 5α-reductase [68,69]. Therefore, the brain is capable of synthesizing various steroid hormones, and their derivatives are known as neurosteroids, including pregnenolone (PREG) and PREG sulfate (PREGS), dehydroepiandrosterone (DHEA) and DHEA sulfate (DHEAS), progesterone, 3α,5α-tetrahydroprogesterone (THP), 3α,5α-tetrahydrodeoxycorticosterone (THDOC), estrogen, and testosterone [70]. CS serves as a substrate for the synthesis of PREGS and DHEAS, directly regulating their intracellular synthesis levels [71]. PREGS and DHEAS play roles in sedation, hypnosis, anxiolysis, and anticonvulsant effects by antagonizing γ-aminobutyric acid (GABA) A receptors [72], which are widely distributed throughout the brain and are the primary inhibitory neurotransmitter receptors in the nervous system. Additionally, DHEA, DHEAS, and PREGS can excite N-methyl-D-aspartate (NMDA) receptors, which are the principal excitatory neurotransmitter receptors involved in memory and learning processes [73]. They increase neuronal excitability and synaptic plasticity, promote wakefulness, and enhance memory and cognitive function [74,75]. Integrative research based on proteomics, metabolomics, and genome-wide association studies (GWASs) has revealed that the levels of CS in individuals with schizophrenia differ significantly from those in healthy controls [76].

## 4. CS and Diseases

Current research has focused on the implications of CS in diseases such as X-linked ichthyosis, diabetes, Alzheimer’s disease, ulcerative colitis, bone metabolic diseases, and cancer (Figure 3), and advancements have been made in understanding the pathophysiological effects and therapeutic applications of CS.

### 4.1. X-Linked Ichthyosis

The protective function of the skin is primarily maintained by the stratum corneum, which consists of multiple layers of lipid matrices and embedded keratinocytes. CS is desulfated into cholesterol by SSase, a process essential for maintaining the homeostasis of stratum corneum lipids. SSase is encoded by the *STS* gene, which is located on the human X chromosome. Mutations in the *STS* gene lead to reduced SSase activity, which impairs the desulfation of CS into cholesterol. This results in an elevated CS/cholesterol ratio, causing the development of recessive X-linked ichthyosis (RXLI) [77].

Excess CS can also inhibit hydroxymethylglutaryl-CoA (HMG-CoA) reductase [60], a rate-limiting enzyme for cholesterol synthesis, resulting in an approximately 50% reduction in cholesterol content in RXLI patients. Recent studies analyzing skin samples from patients with RXLI have revealed a decreased expression of transglutaminase 1 enzyme (TGase 1), an enzyme crucial for the formation of the cornified envelope (CE) and the cornified lipid envelope (CLE). Supraphysiological levels of CS severely interfere with involucrin cross-linking and ω-hydroxyceramide esterification by TGase 1 [20]. Further transcriptomic analysis of *STS*-deficient keratinocytes reveals decreased CS production, along with the dysregulation of epidermal differentiation and lipid metabolism pathways. Notably, there was a marked downregulation in aldehyde dehydrogenase (ALDH) isoforms (ALDH1A1 and ALDH3A1) and the oxytocin receptor (OXTR) [78]. ALDH1A1 and ALDH3A1 are known to play roles in maintaining corneal transparency [79], while OXTR downregulation associates with neurobehavioral abnormalities via impaired oxytocin signaling [80]. Consistent with these findings, RXLI has long been associated with an increased risk of corneal opacities and autism-like features [81]. These findings not only elucidate the pathophysiological cascade from *STS* mutations to RXLI multisystem manifestations but also establish CS as a critical metabolic regulator interfacing epidermal differentiation, neuroendocrine signaling, and ocular homeostasis.

### 4.2. Diabetes

Type 2 diabetes (T2D) is characterized by the progressive deterioration of pancreatic β-cell function and a decrease in β-cell mass. Recent studies have indicated that improvements in metabolic status at the early stages of diabetes can maintain or even reverse β-cell function [82,83]. CS regulates cholesterol homeostasis by targeting key enzymes in the cholesterol synthesis pathway, such as 3-hydroxy-3-methylglutaryl-CoA reductase (HMGCR) and lecithin-cholesterol acyltransferase [60]. Cholesterol metabolism is closely related to pancreatic β-cell function, as the accumulation of cholesterol within cells impairs β-cell function [84]. CS mitigates diabetes in streptozotocin-induced diabetic mice by increasing β-cell mass and function [6]. Additionally, CS regulates β-cell mitochondrial efficiency, reduces the production of reactive oxygen species (ROS), enhances β-cell antioxidant capacity, and decreases apoptosis [6].

CS is a natural ligand for RORα and can regulate various metabolic pathways through RORα [27]. RORα plays a role in controlling blood glucose homeostasis and the development of diabetes by modulating gluconeogenesis [85] and is also closely associated with lipogenesis, insulin production, and insulin sensitivity [86,87]. As a natural ligand of RORα, CS holds significant potential in influencing the pathogenesis and progression of diabetes.

### 4.3. Alzheimer’s Disease

Alzheimer’s disease (AD), a progressive neurodegenerative disorder characterized by cognitive decline and neuronal loss, represents the most prevalent form of dementia worldwide. Epidemiological studies project a dramatic increase in the disease burden, with dementia prevalence expected to double in Europe and triple globally by 2050 [88]. In China, a population-based cohort study (n = 626,276) revealed an AD prevalence of 3.48% and an annual incidence rate of 7.90 per 1000 person-years among older adults [89]. The pathological hallmarks of AD include the extracellular deposition of abnormal β-amyloid protein (Aβ) plaques, resulting from abnormal protein aggregation, and the intracellular accumulation of hyperphosphorylated tau protein (pTau), forming neurofibrillary tangles (NFTs) [90]. Notably, emerging evidence implicates astrocyte–neuron metabolic coupling in AD pathogenesis. Astrocytic SULT2B1b catalyzes the conversion of cholesterol into CS, which is subsequently transported to the mature neuron via the ATP-binding cassette transporter-2 (ABCA2) [91]. CS plays dual pathological roles in AD progression, not only promoting Aβ plaque formation but also facilitating tau hyperphosphorylation and NFT assembly [92,93] (Figure 4).

Molecular dynamics simulations demonstrate that CS exhibits remarkable nucleation-enhancing capacity, accelerating Aβ-42 fibril formation with a 260-fold greater efficiency than cholesterol [94]. Aβ is generated through the sequential proteolysis of amyloid precursor protein (APP) by various proteases, including α-, β-, γ-, η-, and δ-secretases, resulting mainly in the Aβ1-40 and Aβ1-42 forms [95]. APP is a cholesterol-sensitive protein expressed selectively in astrocytes and neurons [96]. Cholesterol and its metabolites can bind to APP, promoting its cleavage into aggregation-prone Aβ1-42, which then associates with cholesterol (in a 1:1 stoichiometry) to form neurotoxic plaques [97]. Compared with cholesterol, CS binds more tightly to Aβ peptides, and when CS accumulates in cell membrane plaques at approximately 50 times the normal level, it significantly accelerates the aggregation of Aβ into toxic, insoluble plaques in the brain, further inducing tau protein aggregation [92]. Biochemical assays show CS differentially activates PKC isoforms, inducing 12.0-, 6.0-, 5.0-, 1.5-, and 1.2-fold increases in PKCε, η, α, δ, and ζ activities, respectively [93,98]. Of particular relevance, PKCα and PKCδ directly phosphorylate tau protein [99]. Thus, CS exacerbates tau protein hyperphosphorylation and tau fibril formation through PKC activation. Therefore, research targeting CS and its rate-limiting enzyme SULT2B1b may provide insights into the metabolic processes and molecular mechanisms underlying AD, potentially identifying new strategies for reversing AD symptoms in the future (Figure 4). Furthermore, CS is a substrate for the synthesis of neuroactive steroids, such as PREGS and DHEAS, which can inhibit GABA_A receptor activation, leading to mood disorders such as depression, anxiety, and epilepsy [100,101].

### 4.4. Ulcerative Colitis

Ulcerative colitis (UC) is a chronic, nonspecific inflammatory bowel disease (IBD) characterized by periods of remission and relapse, involving persistent and confluent inflammatory responses in the mucosa and submucosa of the colon and rectum. It is a lifelong condition with an increasing incidence rate each year [102]. The pathophysiology of UC is complex [103]. Following damage to the intestinal epithelial barrier, macrophages and antigen-presenting cells are activated, leading to the maturation of infiltrating monocytes into macrophages that produce various proinflammatory cytokines, such as TNF, IL-12, IL-23, and IL-6. These cytokines increase the levels of chemokines that attract neutrophils, thereby promoting the development of intestinal inflammation [104]. Nonspecific anti-inflammatory drugs, such as 5-aminosalicylic acid derivatives, corticosteroids, and immunosuppressants, aim to suppress epithelial barrier damage through anti-inflammatory and immunosuppressive effects [103,105]. Recent studies have shown that CS accelerates mucosal healing in UC by promoting cholesterol synthesis in colon epithelial cells [106]. In UC patients, the expression of *SULT2B1* and genes involved in cholesterol biosynthesis is upregulated, and the levels of CS and cholesterol are significantly elevated [106]. Mechanistically, its proinflammatory factors induce the expression of *SULT2B1*, which increases CS synthesis. CS can bind to the Niemann–Pick disease type C2 (NPC2) protein, disrupting cholesterol transport in lysosomes [107,108]. This activates sterol regulatory element-binding protein 2 (SREBP2), which promotes cholesterol biosynthesis and increases intracellular cholesterol levels, ultimately aiding in the repair of damaged intestinal epithelial cells [106]. Additionally, CS has been shown to limit neutrophil recruitment during mucosal damage by inhibiting DOCK2-mediated Rac activation, reducing excessive immune cell accumulation, preventing excessive intestinal inflammation, and accelerating tissue repair [109]. Dietary supplementation with CS can alleviate dextran sulfate sodium (DSS)-induced colitis, providing a novel strategy for UC treatment.

### 4.5. Bone Metabolic Diseases

Bone tissue constantly undergoes remodeling to repair damage and maintain balance through the actions of bone cells, including osteoclast-mediated bone resorption and osteoblast-mediated bone formation [110]. CS has been suggested to play a beneficial role in bone metabolism [111]. In vitro experiments have demonstrated that CS reduces the expression of TRAP (tartrate-resistant acid phosphatase), cathepsin K, and calcitonin receptor in a dose-dependent manner [112]. These factors are critical for osteoclast differentiation and bone resorption. CS has also been reported to activate the AMPK-Sirt1 axis in a *RORα*-independent manner, leading to the inhibition of NFATc1 activity and subsequent suppression of osteoclast differentiation [113]. In inflammation-induced bone destruction mouse models, CS has been shown to increase femoral bone density, trabecular bone number, bone surface area, and bone volume, while significantly reducing trabecular separation [114]. Rheumatoid arthritis (RA) is a chronic inflammatory arthritis characterized by progressive joint destruction [115]. CD4+ T (Th17) cells, which produce IL-17, play crucial roles in RA onset and progression [116]. CS, a natural ligand for RORα, can inhibit the proliferation of Th17 cells and reduce IL-17 production by modulating transcriptional sites in immature T cells. This effect is associated with a decreased expression of glycolytic molecules and p53 [46]. Collectively, these results suggest that CS may serve as a therapeutic strategy in diseases associated with abnormal bone metabolism.

### 4.6. Cancer

CS inhibits T lymphocyte immune function through various mechanisms, facilitating immune evasion by tumor cells (Figure 5). (1) CS suppresses DOCK2-mediated RAC activation and lymphocyte migration. DOCK2, an essential RAC activator for lymphocyte migration and activation, loses its ability to activate Rac upon binding with CS, thereby impeding lymphocyte migration and activation, resulting in severe immune deficiency [117]. Certain tumors, such as colorectal cancer and prostate cancer, produce large amounts of CS, creating a microenvironment that excludes T lymphocytes and resists immunotherapy. Clinical studies have shown a negative correlation between CS levels and the extent of CD8+ T-cell infiltration in human colorectal cancer tissues [118]. In hepatocellular carcinoma, CS inhibits the DOCK2 activity in T cells, promoting CD8+ T-cell exhaustion [119]. High expression of the CS-synthesizing enzyme SULT2B1 in colorectal cancer is associated with poor prognosis [120]. In mouse models, cancer cells that produce CS exhibit resistance to immune checkpoint blockade and antigen-specific T-cell-mediated tumor eradication [118]. (2) CS competitively binds to the TCR. The transmembrane domain of TCR can bind to both cholesterol and CS, but the affinity of CS for the site is 300-fold greater than that of cholesterol. CS binding inhibits TCR dimerization and thereby prevents the formation of the MHC: TCR complex, impairing the infiltration of effector T cells into tumor tissues and thereby leading to tumor immune evasion [117]. (3) CS protects cancer cells from cytotoxic T-cell attacks by modulating the tumor necrosis factor (TNF) receptor signaling pathway. Normally, upon binding with the ligand TNFα, TNF receptor 1 (TNFR1) recruits intracellular adaptor proteins with death domains, such as Fas-associated death domain protein (FADD) and TNFR1-associated death domain protein (TRADD), forming a “death-inducing signaling complex” that initiates caspase cascades (caspase-8, caspase-10, and caspase-3), leading to cytoskeletal proteolysis and apoptosis [121]. When the cellular CS concentration increases, TNFα binding instead mediates the recruitment of the inhibitor of nuclear factor-*κ*B (IκB)-kinase (IKK) complex to TNFR1, thus allowing the activation of the pro-survival nuclear factor-*κ*B (NF-κB) response and promoting cell survival [122]. Thus, CS acts as a critical signal shifting cells from apoptosis to survival. The exact mechanism of this shift is not fully understood but may involve changes in the lipid raft microdomains of cholesterol and sphingolipids, which are crucial for proper TNFR1 function [123].

In addition, CS promotes cancer cell metastasis. Malignant cells shed from the primary tumor are remodeled to survive in the circulation and colonize in distant organs. Once inoculated, the cells must adapt to the available reservoir of new extracellular metabolites and survive under immune surveillance [124]. Metabolomic analysis revealed that CS is highly upregulated in spontaneous lung metastases compared with paired primary tumors [125]. CS, together with cholesterol, phospholipids, and ceramides, is an essential component that maintains the cytoplasmic membrane homeostasis. CS can alter the activity of serine proteases of the coagulation cascade [34], potentially facilitating platelet interactions with circulating cells, including the dissemination of tumor cells in the blood. CS also accelerates the proteolytic hydrolytic activity of matrix metalloproteinase-7 against selective substrates in the extracellular matrix, which may contribute to tumor cell transfer, and may modulate protein kinase C, which is involved in cellular differentiation and oncogenesis [126]. The presence of CS in lung lesions might also be caused by its uptake from the extracellular environment, as well as its production by cancer cells [125]. For the latter hypothesis, a fraction of the cells within the primary tumors may increase the expression of SULT2B1b, which promotes the production of CS, thus supporting the hematogenous spread of metastatic cancer cells to the lungs.

### 4.7. Atherosclerosis

Atherosclerosis is increasingly being reinterpreted as a systemic disorder of sulfate metabolism driven by CS deficiency, challenging traditional views that have long centered on lipid accumulation or inflammation [127]. CS, an amphipathic molecule, enhances cholesterol solubility and facilitates its transport. It also collaborates with sulfated glycosaminoglycans (sGAGs) to maintain the negative charge of the endothelial glycocalyx, which is critical for forming structured water layers that reduce erythrocyte adhesion and ensure proper capillary flow [128]. Atherosclerotic plaques are now proposed to act as compensatory reservoirs, recruiting macrophages and platelets to oxidize LDL and homocysteine for sulfate generation and CS synthesis, albeit at the cost of oxidative damage [129].

Environmental factors play a significant role in modulating sulfate availability. Sunlight exposure, for instance, enhances CS synthesis via endothelial nitric oxide synthase (eNOS)-mediated pathways, while toxins, such as aluminum and glyphosate, disrupt cytochrome P450 enzymes, thereby impairing sulfate metabolism [130]. Dietary enrichment with sulfur-containing compounds, such as those found in garlic, and reduced exposure to toxins may help restore sulfate homeostasis. Conversely, conventional anti-inflammatory therapies, which often suppress compensatory pathways, may inadvertently exacerbate CS deficiency [131]. This emerging paradigm positions CS deficiency as a unifying mechanism underlying atherosclerosis. It highlights the need for in vivo validation to translate these insights into effective therapeutic strategies, potentially offering a novel approach to managing this complex disease.

### 4.8. Lead Poisoning

Lead (Pb) is a harmful heavy metal widely present in the environment and industry that causes dysfunction in multiple organs, particularly affecting the brains of infants, who are relatively susceptible to lead-induced neurotoxicity [132,133]. In the brain, astrocyte cells (CTX) are the primary source of cholesterol production, supplying 98% of the brain’s cholesterol. This not only ensures the structural stability of these cells but also supports neuronal development [134]. However, exposure to Pb disrupts this delicate balance, leading to a decrease in intracellular cholesterol levels and an increase in cell apoptosis [135]. The brain-derived neurotrophic factor (BDNF) is a crucial neurotrophic factor that plays a vital role in neurogenesis, neuronal survival, synaptic plasticity, and memory formation [136,137]. Recent research has shown that pretreating cells with CS can effectively counteract Pb-induced cholesterol loss and cell apoptosis in CTX cells by activating the BDNF signaling pathway [138]. This activation of the BDNF pathway promotes the cleavage of SREBP2 in Schwann cells, which in turn supports the survival of dorsal root ganglia (DRG) neurons by orchestrating cholesterol metabolism [139]. Furthermore, CS treatment has been found to stabilize intracellular Pb levels, indicating that CS does not promote Pb excretion but instead activates the BDNF/TrkB signaling pathway. This activation mediates downstream cholesterol metabolism, compensating for cholesterol loss induced by Pb exposure [140]. However, further research, particularly in vivo studies, is needed to better understand these mechanisms and their potential therapeutic applications.

## 5. CS Quantification Strategies

The comprehensive analysis of CS remains methodologically challenging despite its critical role in metabolism and cellular signaling. Recently, Sanchez et al. performed an in-depth comparative analysis of the analytical strategies for cholesterol and oxysterol sulfates, revealing critical bottlenecks in current methodologies [5]. Current approaches typically involve liquid–liquid extraction (LLE) combined with solid-phase extraction (SPE) for sample purification, with detection achieved through liquid chromatography-tandem mass spectrometry (LC-MS/MS) in negative ion mode [5]. However, the accuracy of CS quantification is intrinsically linked to standardized experimental workflows, including sample collection, storage, extraction, fractionation, separation, detection, and quantification steps. First, sample pretreatment strategies are crucial in the discovery and validation of lipid-based markers. The choice of sample collection tubes, freeze–thaw cycles, and storage conditions can significantly affect the stability of samples and the recovery of plasma lipids [141,142]. For example, plasma oxysterol levels collected with K2-EDTA and citrate collection tubes differed from those in serum samples, supporting the use of EDTA-collection tubes [143]. In cases where serum samples were used, the addition of an antioxidant like butylated hydroxytoluene was suggested to increase the stability of oxysterols [143]. Second, the extraction of steroid-related compounds is typically conducted using LLE protocols, followed by SPE cartridges. LLE protocols are popular due to their simplicity, cost, and efficiency. However, the choice of solvent system can greatly impact the extraction performance, particularly for less abundant lipids. For oxysterol sulfates, extraction by protein precipitation with acetonitrile-ZnSO4, followed by C18 SPE fractionation, resulted in complete recovery [144]. Finally, the accuracy of quantification is affected by the detection sensitivity. In mass spectrometry, oxysterols are usually detected in the positive ion mode, while the presence of the sulfate group facilitates the detection of oxysterol sulfates in the negative ion mode [145]. Targeted detection methods like multiple reaction monitoring (MRM) are often used due to their increased sensitivity and specificity [145]. In chromatographic separation, the hydroxy group position affects the hydrophobicity of oxysterols, influencing chromatographic separation under reverse-phase conditions. This also affects the extraction efficiency of oxysterol sulfates from biological matrices. Despite methodological advancements, reported physiological CS concentrations remain orders of magnitude below biologically active thresholds in in vitro systems (μM range) [144,146]. This discrepancy underscores the urgent need for standardized protocols and ultrasensitive detection platforms.

## 6. Conclusions and Perspectives

Advances in analytical techniques, such as LC-MS and selected ion monitoring, have significantly enhanced our ability to determine CS in various biological materials and elucidate its physiologic roles. CS functions as a regulatory molecule involved in a wide range of biological processes, including keratinocyte differentiation, glucolipid metabolism, bone metabolism, the inflammatory response, the gut microbiota, and neurosteroid synthesis. It has also demonstrated therapeutic efficacy against ulcerative colitis and osteoclast differentiation. One of the most exciting recent findings has been the discovery of immune regulatory functions of CS, which inhibit T lymphocyte activity. This finding suggests that CS could serve as a therapeutic agent for alleviating T-cell-mediated hypersensitivity and as a potential target for treating autoimmune diseases [52]. However, these effects are mostly based on rodent studies. Owing to considerable differences in the biology of cholesterol between humans and commonly used laboratory animals, particularly regarding sulfate synthesis and sulfotransferase specificity, data from rodent models should be critically scrutinized. Indeed, the production of steroid sulfate is almost negligible in rodents compared to primates [147].

CS also exhibits significant pathological damage in certain diseases. For instance, mutations in the *STS* gene lead to reduced activity of SSase, preventing the effective desulfation of CS into cholesterol. This results in the abnormal accumulation in the stratum corneum [78]. Excess CS disrupts the balance of stratum corneum lipid metabolism by enhancing intercellular adhesion, inhibiting HMG-CoA reductase (which reduces cholesterol synthesis), and interfering with the function of TGase 1. These disruptions lead to dry skin and scaly lesions [20]. In cancer, CS weakens T-cell migration by inhibiting the DOCK2/RAC pathway, hinders MHC antigen presentation by competitively binding to the TCR, and modulates TNFR signaling pathways (activating NF-κB rather than the apoptotic pathway). These actions promote tumor immune evasion and cell survival [94]. Additionally, CS accelerates the degradation of the extracellular matrix and hematogenous metastasis of tumors by enhancing the activity of matrix metalloproteinase-7 and the coagulation cascade [148]. These mechanisms highlight the pleiotropic pathological effects of CS and provide key insights for targeted therapeutic interventions.

CS is biosynthesized by the sulfonation of cholesterol through the sulfotransferase SULT2B1b. Thus, deciphering the regulation of SULT2B1 expression, such as *SULT2B1* transcription and tissue distribution, may enable us to understand the physiologic significance of CS. In addition to being detected in the skin, SULT2B1b has also been extensively detected in the brain, lower intestine, lung, tonsil, platelets, testes, and ovaries [149]. Further studies are needed to explore the definitive biological basis for the expression in each specific tissue. Although primarily responsible for the sulfonation of cholesterol, SULT2B1b has also been shown to sulfate oxysterols, including oxygenated derivatives of cholesterol (e.g., 7-ketocholesterol, 24- or 25-hydroxycholesterol, and epoxycholesterol) and hydroxysteroids (e.g., DHEA, androstenediol, and PREG) [93]. Therefore, continued efforts need to be made in the system regulation of the sulfation of cholesterol and other oxysterols, particularly the implications of these sulfated molecules and their homeostasis.

## Figures and Tables

**Figure 1 biomolecules-15-00646-f001:**
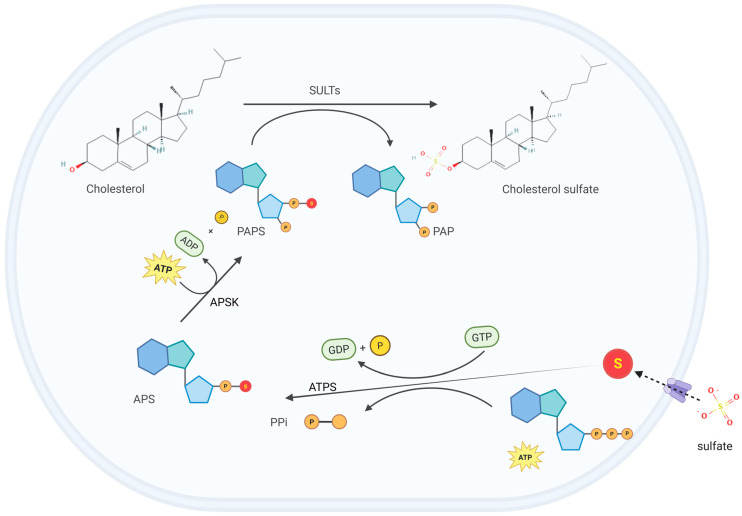
Cholesterol sulfate (CS) biosynthesis. Dietary-derived sulfate is transported into cells through specific transporters. Once inside the cell, the adenosine monophosphate (AMP) moiety of adenosine triphosphate (ATP) is transferred to sulfate by the ATP sulfurylase (ATPS) activity within the bifunctional 3′-phosphoadenosine 5′-phosphosulfate (PAPS) synthase complex, generating adenosine-5′-phosphosulfate (APS). Subsequently, the APS kinase (APSK) domain of PAPS synthase phosphorylates APS at the 3′-position to generate PAPS, which contains the active sulfate group. Finally, sulfotransferases (SULTs) catalyze the transfer of the active sulfate group from PAPS to the hydroxyl group of cholesterol, producing CS. This figure was created in Biorender. Yu, X. (2025) https://BioRender.com/itac2bb (accessed on 10 March 2025).

**Figure 2 biomolecules-15-00646-f002:**
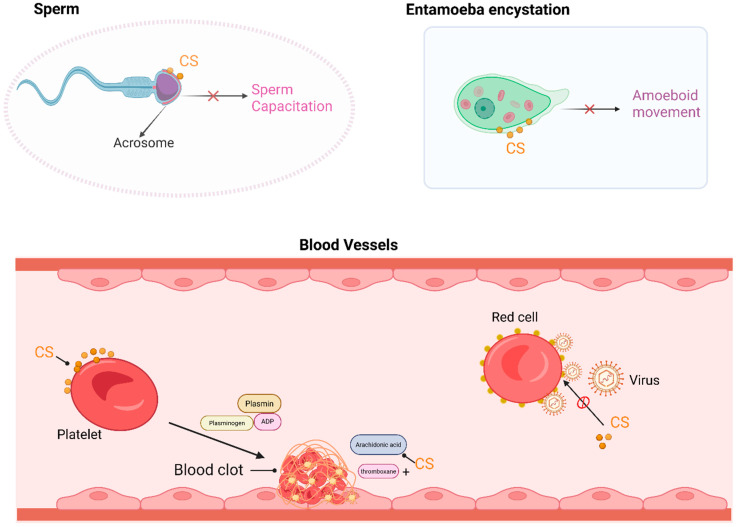
CS functions as a component in biological membranes. In the sperm plasma membrane, primarily in the acrosomal region, the hydrolysis of CS can lead to membrane destabilization and trigger motility events. In Entamoeba encystation, CS dose-dependently induces and maintains encysting cells as spherical maturing cysts, with almost no phagocytotic activity. In platelets, CS binds to the membrane and enhances ADP (Adenosine diphosphate)- and thrombin-induced platelet aggregation and serotonin secretion. CS is also a potent inhibitor of Sendai virus fusion to erythrocyte membranes. This figure was created in Biorender. Yu, X. (2025) https://BioRender.com/oaeglkg (accessed on 10 March 2025).

**Figure 3 biomolecules-15-00646-f003:**
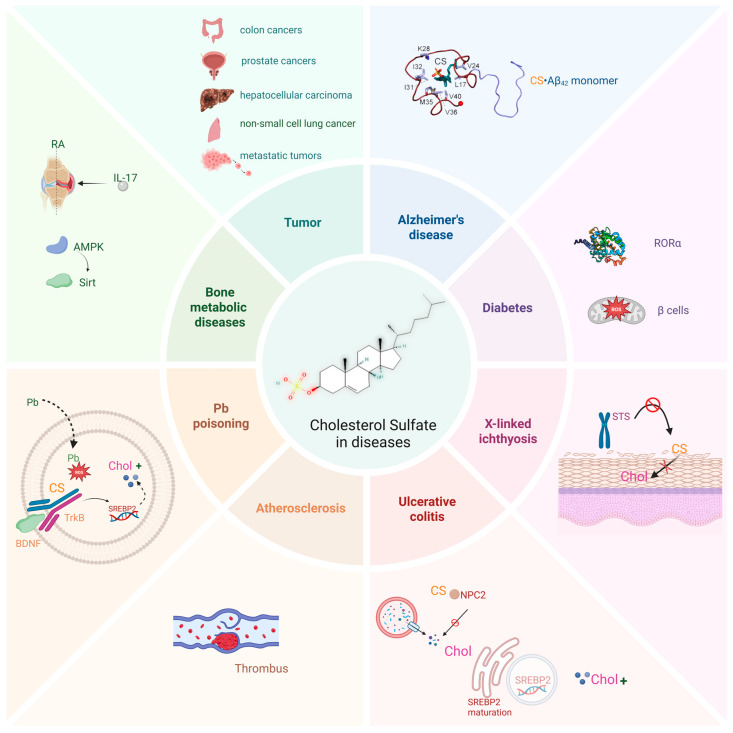
CS is implicated in various diseases. CS plays an important biological role in various diseases, such as different types of cancers, Alzheimer’s disease, Atherosclerosis, diabetes, X-linked ichthyosis, and ulcerative colitis. This figure was created in Biorender. Yu, X. (2025) https://BioRender.com/ls81p2c (accessed on 10 March 2025).

**Figure 4 biomolecules-15-00646-f004:**
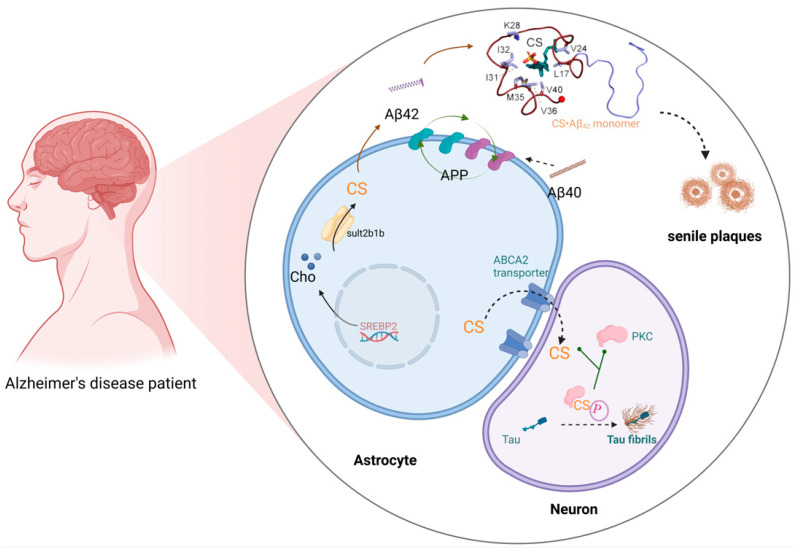
CS in Alzheimer’s disease. CS binds to amyloid precursor protein (APP) and promotes the cleavage of APP into aggregation-prone Aβ42, fostering the formation of Aβ plaques. Sult2b1b in the astrocyte cytosol converts cholesterol into CS, both of which are provided by the astrocyte to the mature neuron via the ATP-binding cassette transporter-2 (ABCA2), which does not synthesize cholesterol. Once in the neuron, CS binds protein kinase C (PKC), causing it to hyperphosphorylate tau protein fibrils, which aggregate into toxic neurofibrillary tangles. This figure was created in Biorender. Yu, X. (2025) https://BioRender.com/7dts6k3 (accessed on 10 March 2025).

**Figure 5 biomolecules-15-00646-f005:**
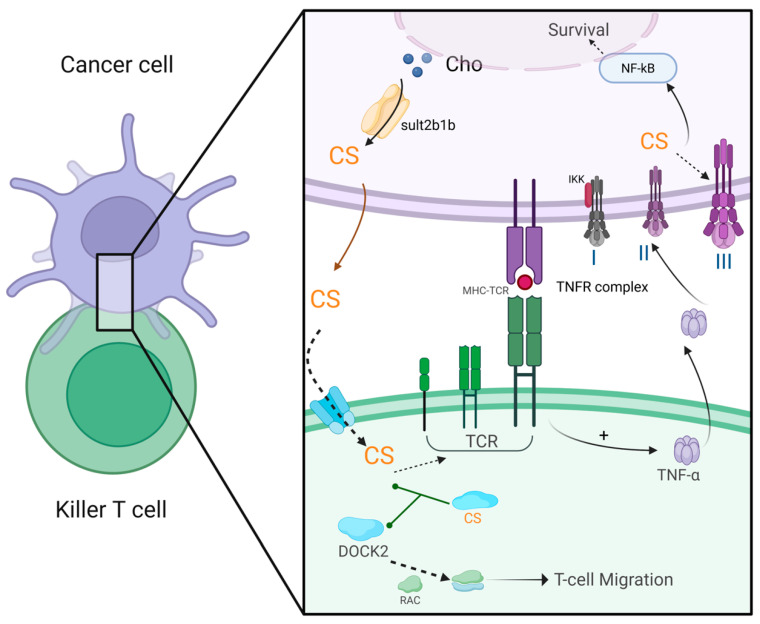
CS in cancer biology. In cancer cells, Sult2b1b synthesizes CS, which is exported to the extracellular space via transporters. Upon entering mature killer T cells, CS binds to DOCK-2, blocking its ability to activate Ras-related C3 botulinum toxin substrate GTPase (RAC). This inhibition traps RAC in an inactive GDP-bound state, impairing T-cell migration and tumor infiltration. Simultaneously, CS binds the transmembrane domain of the TCR, preventing its dimerization and disrupting the formation of functional TCR-MHC complexes, thereby suppressing antigen recognition and cytokine release (e.g., TNFα). Within the cancer cell, elevated CS levels redirect TNFα signaling the following: instead of promoting apoptosis via FADD/caspase-8 activation (Complex II), TNFα binding stabilizes a pro-survival TNFR complex (Complex III). The pro-survival pathway is initiated by the activation of IKK, which phosphorylates IκB, causing release of NF-κB and exposing its nuclear-localization signaling peptide. As a result, NF-κB translocates to the nucleus and alters gene expression to inhibit apoptosis and promote survival. This figure was created in Biorender. Yu, X. (2025) https://BioRender.com/2bm0ov0 (accessed on 10 March 2025).

## Data Availability

No new data were created or analyzed in this study.

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
