# Peer review of "Cholesterol Sulfate: Pathophysiological Implications and Potential Therapeutics"

_biomolecules, 2025, doi:10.3390/biom15050646_

Round 1

Reviewer 1 Report

Comments and Suggestions for Authors

In this paper the authors would aim at providing an overview of the literature reporting the pathophysiological involvement of cholesterol sulfate (CS) in several human diseases and the potential therapeutic outcomes. Despite the relevance of the topic, the present version of the manuscript needs an in-depth revision since some aspects, necessary for an exhaustive dissertation of this topic, have not been fully and clearly developed.

MAJOR POINTS

  • Lacks are particular evident in Section 4 and in the related Figures 3, 4, and 5, as detailed hereafter:
  • It is not clear the meaning of the following sentences (lines 348-358, it is not clear the connection with CS; lines 378-381, it is not clear the connection with diabetes; lines 532-536, it is not clear the connection with lead poisoning). They must be clarified. Addition of further references could be useful at this scope (see also major point 5);
  • 3,4, and 5 do not exactly depict what is reported in the text. They show some elements that have not been considered in the manuscript or are explained in other sections, thus making some confusion (in fig. 3, vitamin D and thrombus, signaling in Pb poisoning, signaling in rheumatoid arthritis, and HNF4α acetylation in diabetes; in fig. 4, signaling in astrocyte, including SREBP2 and ABCA2). The authors have to briefly and properly illustrate these factors in the text or otherwise delete them. According to that, also the legends should describe more in detail the figure content (in particular fig. 5 does not clearly depicts what is reported in its legend).

  • According to the abstract, the review would aim to address the potentiality of therapies approaching CS and CS-dependent signaling for disease treatment. Nevertheless, apart from a few elusive author considerations, this aspect has not been properly discussed. I would recommend the authors to better treat it in section 4 and report evidences about the efficacy of CS-targeting therapies, if available.

  • In Conclusions, the authors mention techniques recently employed for CS determination but they do not have treated this issue before. A brief section concerning the analytical approaches for CS quantification in biological specimens, including the most relevant evidences in disease conditions, should be added (at this purpose, see also lines 41-46 and Ref. 5, they should be improved). Moreover, as reported in section 4, CS appears also negatively involved in some pathologies (e.g. X-linked ichtyosis, cancer): this aspect also deserves to be commented in Conclusions by the authors.

  • Some sentences are not correct or need more in-depth explanation:
  • Lines 222-224, according to Ref. 37, CS inhibits and does not stimulate pro-inflammatory cytokines’ production;
  • Line 257, CS is not encoded by Sult2b1 gene but it is a substrate of the encoded enzyme Sult2b1;
  • Lines 258-259, add more details on Refs. 46 and 49 content;

  • the following References seem improper, please check them and replace them with more relevant ones or add further Refs; moreover, references lack in other sentences:
  • 3, line 40, it does not focus on androsterone, cholesterol, and pregnenolone;
  • 13 and 14, lines 99-102, ref. 13 only refers to breast cancer while ref 14 does not seem talk about cancer;
  • 19, lines 129-131, there are no evidences about AP-1;
  • 22, lines 142-145;
  • 26, lines 166-168, replace it with more recent and pertinent Refs.;
  • Lines 178-181, add Ref.;
  • Lines 212-217, add Ref.;
  • Lines 253-255, Ref. 48 in the lines above actually concerns eye but there aren’t examples for brain and uterus;
  • Lines 255-258, add Ref.;
  • Refs 51 and 52, lines 267-269;
  • 54, lines 271-275;
  • 63, lines 305-307;
  • Lines 311-317, add Ref.;
  • Refs 65 and 66, lines 322-326;
  • Lines 368-371, add Ref.;
  • Lines 378-380, add Ref.;
  • Lines 380-381, add Ref.;
  • 80-82, lines 381-383;
  • 83, lines 387-388;
  • Lines 392-394, add Ref.;
  • 86 lines 411-413, it is an old study on keratinocytes, replace it or otherwise explain its relevance in the 4.3. context;
  • 63, lines 406-408;
  • Lines 392-394, add Ref.;
  • 88-91, lines 418-420;
  • 93, lines 425-430;
  • Lines 433-436, add Ref.;
  • 96, lines 438-440, it reports that cholesterol sulfate binds to NPC2 but there are no data about cholesterol content in lysosomes and endoplasmic reticulum;
  • Lines 440-442, add Ref.;
  • Lines 442-447, add Ref.;
  • 98, lines 449-451;
  • 101, lines 455-457;
  • 102, lines 457-460;
  • 108 and 109, lines 479-480;
  • 110, lines 480-482;
  • 111, lines 493-496, further refs. to prove TNFα/NF-kB pathway modulation by CS are needed;
  • 112, lines 497-499;
  • 20, lines 520-521;
  • Lines 528-531, add Ref.;
  • 120 and 121, lines 532-534.

MINOR POINTS

  • Abbreviations, please check that all of them are defined the first time they appear (abstract and main text). Moreover, they have to been explained also in the legend of the first figure they appear.
  • Please check that in the References list all information is present (e.g. journal name, …).
  • Please check punctuation (e.g. spaces before Ref. quotation, commas).
  • Please write all gene names in italic.
  • 1, please add AS and APSK near the arrows of the corresponding enzymatic reactions.
  • Line 322, DHEAS instead of DHEA?

Overall, in my opinion, a major, substantial revision of the manuscript is required to improve its quality and allow its publication.

Author Response

MAJOR POINTS

  1. Lacks are particular evident in Section 4 and in the related Figures 3, 4, and 5, as detailed hereafter:

It is not clear the meaning of the following sentences (lines 348-358, it is not clear the connection with CS; lines 378-381, it is not clear the connection with diabetes; lines 532-536, it is not clear the connection with lead poisoning). They must be clarified. Addition of further references could be useful at this scope (see also major point 5);

3,4, and 5 do not exactly depict what is reported in the text. They show some elements that have not been considered in the manuscript or are explained in other sections, thus making some confusion (in fig. 3, vitamin D and thrombus, signaling in Pb poisoning, signaling in rheumatoid arthritis, and HNF4α acetylation in diabetes; in fig. 4, signaling in astrocyte, including SREBP2 and ABCA2). The authors have to briefly and properly illustrate these factors in the text or otherwise delete them. According to that, also the legends should describe more in detail the figure content (in particular fig. 5 does not clearly depicts what is reported in its legend).

Response: We sincerely thank the reviewer for their thoughtful and constructive comments. We have thoroughly revised our manuscript in accordance with the feedback provided.

Specifically, the sentences on lines 348–358 (Section 4.1), lines 378–381 (Section 4.2), and lines 532–536 (Section 4.8) in the original manuscript have been clarified and are highlighted in yellow in the revised version for ease of review. Additionally, elements relevant to Figures 3, 4, and 5 have been incorporated into the revised manuscript, and the figure legends have been updated to provide more detailed descriptions of the content.

  1. According to the abstract, the review would aim to address the potentiality of therapies approaching CS and CS-dependent signaling for disease treatment. Nevertheless, apart from a few elusive author considerations, this aspect has not been properly discussed. I would recommend the authors to better treat it in section 4 and report evidences about the efficacy of CS-targeting therapies, if available.

Response: We appreciate the reviewer’s insightful comment. Currently, there are limited treatment strategies targeting CS in the context of this disease, and unfortunately, early clinical trials have not yet yielded promising results..

  1. In Conclusions, the authors mention techniques recently employed for CS determination but they do not have treated this issue before. A brief section concerning the analytical approaches for CS quantification in biological specimens, including the most relevant evidences in disease conditions, should be added (at this purpose, see also lines 41-46 and Ref. 5, they should be improved). Moreover, as reported in section 4, CS appears also negatively involved in some pathologies (e.g. X-linked ichtyosis, cancer): this aspect also deserves to be commented in Conclusions by the authors.

Response: We are grateful for the reviewer’s valuable feedback. As suggested, we have added a brief section (Section 5) on analytical approaches for CS quantification in the revised manuscript. Additionally, the negative impact of CS on certain pathologies has been addressed in the revised conclusion, which is highlighted in yellow for clarity.

  1. Some sentences are not correct or need more in-depth explanation:

Lines 222-224, according to Ref. 37, CS inhibits and does not stimulate pro-inflammatory cytokines’ production;

Line 257, CS is not encoded by Sult2b1 gene but it is a substrate of the encoded enzyme Sult2b1;

Lines 258-259, add more details on Refs. 46 and 49 content;

Response: We appreciate the reviewer's insightful comments. As suggested, the concerns raised have been addressed and corrected in the revised manuscript accordingly.

  1. The following References seem improper, please check them and replace them with more relevant ones or add further Refs; moreover, references lack in other sentences:

3, line 40, it does not focus on androsterone, cholesterol, and pregnenolone; 13 and 14, lines 99-102, ref. 13 only refers to breast cancer while ref 14 does not seem talk about cancer; 19, lines 129-131, there are no evidences about AP-1; 22, lines 142-145; 26, lines 166-168, replace it with more recent and pertinent Refs.; Lines 178-181, Lines 212-217, add Ref.; Lines 253-255, Ref. 48 in the lines above actually concerns eye but there aren’t examples for brain and uterus; Lines 255-258, add Ref.; Refs 51 and 52, lines 267-269; 54, lines 271-275;63, lines 305-307;Lines 311-317, add Ref.; Refs 65 and 66, lines 322-326; Lines 368-371, Lines 378-380, Lines 380-381, add Ref.; 80-82, lines 381-383; 83, lines 387-388; Lines 392-394, add Ref.; 86 lines 411-413, it is an old study on keratinocytes, replace it or otherwise explain its relevance in the 4.3. context; 63, lines 406-408; Lines 392-394, add Ref.; 88-91, lines 418-420; 93, lines 425-430; Lines 433-436, add Ref.; 96, lines 438-440, it reports that cholesterol sulfate binds to NPC2 but there are no data about cholesterol content in lysosomes and endoplasmic reticulum; Lines 440-442, Lines 442-447, add Ref.; 98, lines 449-451; 101, lines 455-457; 102, lines 457-460; 108 and 109, lines 479-480; 110, lines 480-482; 111, lines 493-496, further refs. to prove TNFα/NF-kB pathway modulation by CS are needed; 112, lines 497-499; 20, lines 520-521; Lines 528-531, add Ref.; 120 and 121, lines 532-534.

Response:  We sincerely thank the reviewer for their thoughtful and constructive feedback, and we apologize for the mistakes in the references. We have carefully reviewed and corrected the relevant references as pointed out by the reviewer. Thank you very much.

MINOR POINTS

  1. Abbreviations, please check that all of them are defined the first time they appear (abstract and main text). Moreover, they have to been explained also in the legend of the first figure they appear.

Response: As suggested, all abbreviations are now defined upon their first appearance in the text and are also explained in the legend of the first figure they appear in.

  1. Please check that in the References list all information is present (e.g. journal name, …).

Response: The revised version now includes all the necessary information in the references..

  1. Please check punctuation (e.g. spaces before Ref. quotation, commas).

Response: The punctuation has been appropriately revised and corrected throughout the manuscript.

  1. Please write all gene names in italic.

Response: As suggested, all gene names are written in italic.

      5. Fig.1, please add AS and APSK near the arrows of the corresponding enzymatic reactions.

Response: As suggested, AS (ATPS in the revised version) and APSK have been added in the revised Fig. 1.

  1. Line 322, DHEAS instead of DHEA?

Response: DHEAS has been added in the revised version.  

Reviewer 2 Report

Comments and Suggestions for Authors

This manuscript by Yu X et al. provides a comprehensive review of cholesterol sulphate (CS) that is associated with various diseases ranging from diabetes to ulcerative colitis, cancer, bone metabolic, skin and Alzheimer’s disease. The review focus on the biological pathways influencing human health and disease is timely and should capture the wide biological science audience. The references are accurate.

A few concerns should be addressed:

  1. Figures 1 -3 should be revised for the sake of clarity.
  2. What is Virul in Figure 2?

Author Response

1.Figures 1 -3 should be revised for the sake of clarity.

Response: As suggested, the figures have been revised to enhance clarity.

2. What is Virul in Figure 2?

Response: Appoly for this misspelling. It has been replaced by "virus" in the revised Figure.  

Reviewer 3 Report

Comments and Suggestions for Authors

The review is devoted to the cholesterol derivative - cholesterol sulfate, its effect on various systems and functions of the body, the occurrence and course of pathological processes in the body. The topic of the review is undoubtedly relevant and despite numerous studies devoted to the effect of this compound on processes in the body, a lot of ambiguous and unanswered questions are still associated with it. The presented review covers various areas of research related to this compound and its effect on the body. The list of references includes both relevant studies of recent years and significant earlier works. A selective check of the references did not reveal any inconsistencies.

The authors have carried out comprehensive work. The presented material not only helps to navigate the results obtained on this topic and find the necessary reference for narrow specialists. It also guides researchers who came to this topic from related fields of science. The article is rather a Tutorial review, which is in demand in this area of ​​​​research. Therefore, I believe that it should be published after making a number of changes.

  • The review uses a large number of abbreviations for compounds in text and figures, but not all abbreviations are introduced in the text (for example, SULT, ADP). These abbreviations are actively used, but they must be indicated: both according to the rules for formatting articles, and for specialists in related fields. You need to go through the text and Figures and check all the abbreviations.
  • The description of Figure 1 is smeared between the text and the caption to the figure. In both places it is not complete enough and requires additional information. It is impossible to fit it completely into the caption. A short title is needed. But in the text, it is necessary to describe the Figure in more detail. It is precisely possible and necessary to decipher the abbreviations used in the figure.
  • The paragraph from line 90 to 102 looks separate. It needs an additional paragraph linking sections 2 and 3 of the article. Perhaps the authors tried to do this, but it didn't work out very well.
  • Sections 3.1 and 4.1 describe the effect of Cholesterol sulfate on skin lipids and its condition. It is better to provide this information in part 3.1, and in 4.1 provide a link and go to the description of the disease.

Minor

In part of the text there is a space between the text and the bracket from the reference, and in the other half there is not. The text needs to be edited so that the space is everywhere.

Author Response

  1. The review uses a large number of abbreviations for compounds in text and figures, but not all abbreviations are introduced in the text (for example, SULT, ADP). These abbreviations are actively used, but they must be indicated: both according to the rules for formatting articles, and for specialists in related fields. You need to go through the text and Figures and check all the abbreviations.

Response: As suggested, all the abbreviations in text and figures have been introduced.

  1. The description of Figure 1 is smeared between the text and the caption to the figure. In both places it is not complete enough and requires additional information. It is impossible to fit it completely into the caption. A short title is needed. But in the text, it is necessary to describe the Figure in more detail. It is precisely possible and necessary to decipher the abbreviations used in the figure.

Response: We sincerely thank the reviewer for thoughtful and constructive comment. We have now revised our manuscript according to the comments.

  1. The paragraph from line 90 to 102 looks separate. It needs an additional paragraph linking sections 2 and 3 of the article. Perhaps the authors tried to do this, but it didn't work out very well.

Reply: Section 2 delves into the biosynthesis, metabolism, and excretion of CS. Specifically, the paragraph spanning lines 90 to 102 focuses on the metabolic process of CS, highlighting the crucial role of the STS gene, which significantly impacts CS metabolism.

  1. Sections 3.1 and 4.1 describe the effect of Cholesterol sulfate on skin lipids and its condition. It is better to provide this information in part 3.1, and in 4.1 provide a link and go to the description of the disease.

Response: We appreciate the reviewer's insightful comments. To enhance the logical flow, we have revised two sections of the manuscript. Section 3.1 now specifically addresses the physiological and pathological processes involving CS, while Section 4.1 focuses on its implications in various diseases. These modifications aim to provide a clearer and more coherent presentation of the subject matter.

Minor

In part of the text there is a space between the text and the bracket from the reference, and in the other half there is not. The text needs to be edited so that the space is everywhere.

Response: As suggested, a space has been consistently added between the text and the reference bracket throughout the entire manuscript to improve readability.